# ChatGPT as an Information Source for Patients with Migraines: A Qualitative Case Study

**DOI:** 10.3390/healthcare12161594

**Published:** 2024-08-10

**Authors:** Pascal Schütz, Sina Lob, Hiba Chahed, Lisa Dathe, Maren Löwer, Hannah Reiß, Alina Weigel, Joanna Albrecht, Pinar Tokgöz, Christoph Dockweiler

**Affiliations:** Department Digital Health Sciences and Biomedicine, Professorship of Digital Public Health, School of Life Sciences, University of Siegen, 57076 Siegen, Germany; sina.lob@student.uni-siegen.de (S.L.); hiba.chahed@student.uni-siegen.de (H.C.); lisa.dathe@student.uni-siegen.de (L.D.); maren.loewer@student.uni-siegen.de (M.L.); hannah.reiss@student.uni-siegen.de (H.R.); alina.weigel@student.uni-siegen.de (A.W.); joanna.albrecht@uni-siegen.de (J.A.); pinar.tokgoez@uni-siegen.de (P.T.); christoph.dockweiler@uni-siegen.de (C.D.)

**Keywords:** health information, migraine, tension headache, ChatGPT, large language model, semi-structured interview, patient communication, AI in healthcare

## Abstract

Migraines are one of the most common and expensive neurological diseases worldwide. Non-pharmacological and digitally delivered treatment options have long been used in the treatment of migraines. For instance, migraine management tools, online migraine diagnosis or digitally networked patients have been used. Recently, applications of ChatGPT are used in fields of healthcare ranging from identifying potential research topics to assisting professionals in clinical diagnosis and helping patients in managing their health. Despite advances in migraine management, only a minority of patients are adequately informed and treated. It is important to provide these patients with information to help them manage the symptoms and their daily activities. The primary aim of this case study was to examine the appropriateness of ChatGPT to handle symptom descriptions responsibly, suggest supplementary assistance from credible sources, provide valuable perspectives on treatment options, and exhibit potential influences on daily life for patients with migraines. Using a deductive, qualitative study, ten interactions with ChatGPT on different migraine types were analyzed through semi-structured interviews. ChatGPT provided relevant information aligned with common scientific patient resources. Responses were generally intelligible and situationally appropriate, providing personalized insights despite occasional discrepancies in interaction. ChatGPT’s empathetic tone and linguistic clarity encouraged user engagement. However, source citations were found to be inconsistent and, in some cases, not comprehensible, which affected the overall comprehensibility of the information. ChatGPT might be promising for patients seeking information on migraine conditions. Its user-specific responses demonstrate potential benefits over static web-based sources. However, reproducibility and accuracy issues highlight the need for digital health literacy. The findings underscore the necessity for continuously evaluating AI systems and their broader societal implications in health communication.

## 1. Introduction

### 1.1. Impact of Migraines on Individuals

A migraine is defined as a complex, multifactorial, neurovascular dysfunction of the brain [1] that affects approximately 15% of the world’s population [2]. It can be classified as a primary headache disorder and divided into a migraine without an aura and a migraine with an aura. Furthermore, broad distinctions can be made, e.g., between clinical pictures with differentiable symptoms, diagnoses, and therapies [2]. The disease pattern is characterized by persistent symptoms that lead to psychological stress over a more extended, longer period of time and can manifest themselves as reduced enjoyment in life, anxiety, depression or the manifestation of panic before recurring migraine attacks. Consequently, a migraine has an enormous impact on a person’s health and quality of life [3]. It is associated with far-reaching consequences on individual and societal levels, which are responsible for massive losses in the global economy. A migraine is considered one of the most common causes of dysfunction globally in people under 50 [4]. It shows a strong association with the female gender [5]. Thus, the incidence of the disease is 2.5 to 3 times higher in women [6]. For those affected, the chronification of the disease is mainly characterized by a drastic restructuring of their behavior and their way of living, which is caused by various circumstances such as the symptom management, the integration and acceptance of the disease as well as the adequate method of finding information regarding it [7]. In this context, there emerges a broad, ever-evolving, and continually increasing demand for information in disease management.

### 1.2. Role of AI in Health Information

The high diversity and density of information and the heterogeneous quality represent a major challenge for patients searching for evidence-based information that meets their needs. AI has great potential to convey targeted patient information and services, including those relating to the diagnosis, therapy, prevention, and health promotion of migraines, to a broad audience quickly, conveniently, cost-effectively, efficiently, and independently of location and time [8]. Affected people can easily access comprehensive health information, learn about health issues, and receive support to manage health problems using digital channels [9,10], which relate to the wide range of different disease patterns and symptoms of migraines. Furthermore, AI can provide information on public health issues, answer questions on health promotion and disease prevention, understand the impact of social and environmental factors on individual health, or provide information on community health programs [11]. Critical attention must be paid to “infodemics”, which are negatively associated, especially in the public health context. This, in turn, requires that patients have a certain level of health literacy and digital skills [8,12]. To address this challenge, AI in chatbots such as Chat Generative Pretrained Transformer (ChatGPT) could be used as a supporting tool for searching for information regarding individual health.

ChatGPT is an AI-based large language model (LLM) released in November 2022 that has been trained with large amounts of data to respond to multiple languages and generate refined and sophisticated responses based on advanced modeling [13]. ChatGPT is thus capable of performing various natural language process (NLP) tasks. LLMs recognize and learn patterns in the data structure and consider possible stereotypes when reproducing the data [14]. In a study by Shahsavar and Choudhury [15], it was found that about 80% of the respondents considered using ChatGPT for self-diagnosis [15]. The rapid production of large amounts of text, as a result of the use of ChatGPT, may further reinforce the spread of misinformation in the form of an AI-driven infodemic [16].

Despite the potential benefits, there is a notable research gap in evaluating the reliability and effectiveness of AI-generated health information, specifically for migraine patients. This gap highlights the necessity for an investigation into the potential of ChatGPT as a suitable tool for providing trustworthy and evidence-based health information in the context of migraine management [17].

Unlike existing studies that examine the quality of information generated by ChatGPT in the field of cancer information [14], this paper focuses on health information generated by ChatGPT for people with migraines. Given the complexity and prevalence of migraines, as well as the continuous need for reliable health information among patients, it is crucial to investigate whether ChatGPT can serve as a suitable tool for providing trustworthy and evidence-based health information in the management of migraines. The high incidence of migraines, their substantial impact on quality of life, and the diverse information needs of patients justify a focused evaluation of ChatGPT’s effectiveness in this context. Additionally, migraines present unique aspects, such as their episodic yet chronic nature, the wide range of triggers and symptoms, and the significant overlap with psychological conditions like anxiety and depression. These factors create a distinct need for personalized and adaptive information and support, making migraines particularly suitable for investigation with AI-driven tools like ChatGPT. For instance, while chronic diseases such as diabetes and hypertension involve continuous monitoring and consistent medication regimens, migraines require highly individualized management strategies due to their unpredictable nature and variety of triggers. This makes personalized AI responses through ChatGPT particularly valuable for migraine management, more so than for other chronic conditions that follow a more predictable course.

### 1.3. Study Aim and Research Questions

Against this background, the aim of this study is to investigate whether ChatGPT is suitable as an information source for patients with different types of migraines. This study does not seek to replace professional medical consultation but aims to investigate the potential of ChatGPT in supporting patients’ information needs. This analysis is crucial because the quality and accuracy of information can significantly impact patients’ understanding and management of their condition. By using ChatGPT, we aim to understand its potential and limitations as a tool for disseminating health information. Therefore, the focus in this study is to investigate the underlying LLM, without delving into the implications or contexts of clinical applications for healthcare providers or patients. The following research questions guided the study:-Are the recommendations given by ChatGPT qualitatively equivalent to standard patient information (correct in content, understandable in language, appropriate to the situation and comprehensible)?-To what extent are individual requirements due to variations in the course of disease addressed?-How consistent are the answers and recommendations in health counseling for people with migraines?

In the following paper, first, the methodological approach of the analysis is explained. For this, the research object and methods of data analysis are described in detail. Afterwards, the results of the content analysis and a discussion of the insights as well as the limitations of the study are presented.

## 2. Materials and Methods

A deductive, qualitative study design was conducted to address the research questions. The selection of ChatGPT for this study was based on two factors: its advanced language processing capabilities and its widespread use among individuals seeking health information online. In comparison to other AI tools, such as IBM Watson, HealthTap, or Ada Health, ChatGPT offers a unique combination of accessibility, linguistic versatility, and rapid response generation, which makes it a prominent candidate for evaluating AI-driven health information dissemination [18,19].

The conversation with ChatGPT was achieved with semi-structured guided interviews, which could flexibly be adapted to the course of the conversation without excluding relevant topics. The conversations with ChatGPT were carried out by interviewers who had been trained in advance for each case, so that appropriate prior knowledge of the disease was available. Data were collected using ChatGPT version 3.5 from April to May 2023.

### 2.1. Sample and Case Selection

Based on the systematization of Solomon [20], migraine types and complications were identified and ten fictitious cases were designed, each suffering from a migraine type or complication and having typical characteristics (e.g., age, gender, health behavior, migraine course and symptoms). Objects of the investigation were the nine different migraine characteristics, which can be differentiated regarding the classification of the migraine and the partly marginally different disease courses [20], as well as the demarcation to one of the most frequently occurring headache types, the tension headache (Table 1). These theoretically and empirically derived casuistics comprise seven women and three men ranging in age from 13 to 37 years.

### 2.2. Interview Guide and Survey

A structured interview guide was developed for conducting the conversations with ChatGPT, divided into eight themes (Table 2). The guide provided a thematic orientation for the comparability of the case analysis using the same important questions in every case. At the same time, the questions were individualized in accordance with the characteristics of the cases in the wording and the sequence of the questions during the interview, to simulate a real conversation with a user or potential patient. Depending on the situation, the diagnosis-related guiding questions also differed in the cases that had already received a disease diagnosis from their treating physician. Two rounds per case (*n* = 20) were conducted with independent OpenAI accounts to check the reproducibility of the responses.

### 2.3. Data Preparation and Analysis

Data preparation followed the guidelines for structuring content analysis [21]. The content of the interviews was paraphrased, semantically condensed, and assigned to the deductively formed analysis categories of (1) disease definition, (2) disease genesis and treatment, (3) effects of the disease on everyday life and (4) a future perspective. The deductive structuring was based on standard patient information for patients with migraines. The paper by Solomon [20], the patient information of the Migräne Liga e. V. Germany [22] and gesund.bund.de of the German Federal Ministry of Health [23] were chosen as references for data analysis purposes. In addition, the International Classification of Headache Disorders (ICHD-3) was an important basis [24]. The ChatGPT responses were checked for content accuracy within the defined categories independently by two authors (SL und HR).

Furthermore, it was determined whether the information provided by ChatGPT was reproducible. This criterion was fulfilled when comparable information was available in both rounds. The Hamburg comprehensibility model was used to analyze linguistic comprehensibility [25]. This model categorized the texts according to four comprehensibility characteristics: (1) simplicity, (2) structure and order, (3) brevity and conciseness, and (4) stimulating additions. The criterion ‘simplicity’ measures whether the text presents the content in easily readable sentences and with familiar words. ‘Structure and order’ describes whether a common thread is followed and essential information is distinguished from non-essential information. The criterion ‘brevity and conciseness’ measures whether the text presents the information in the necessary length. Furthermore, the criterion ‘stimulating additions’ comprises the text elements have been personalized and have varied formulations [25].

The parameter ‘situational appropriateness’ was examined using three categories: (1) patient-specific expression: the language model responds to the patient’s remarks in a personalized way; (2) empathy: the language model shows understanding and compassion for the patient’s described situation in the course of the conversation and responds emotionally to the concern; and (3) language in general: the language model adapts the choice of language to the seriousness of the situation.

Subsequently, the information given by ChatGPT regarding disease definition, disease genesis and treatment, effects on everyday life and a future perspective was examined. This information is considered adequate if ChatGPT can provide specific sources that exist and match the chat’s content. By asking ChatGPT “Do you have further information?”, traceability was investigated for sources generated from the question, such as websites or other publications, where traceability is accurate if a link is mentioned, no error message is shown, or a short search on the given website shows a hit on the specific migraine form.

## 3. Results

### 3.1. Content Analysis

For the content analysis, the defined categories (disease definition, disease genesis and treatment, impact of the disease on everyday life and a future perspective) were examined. In the area of disease definition, ChatGPT names one or more potential diseases for diagnosis in each round which could fit the symptoms mentioned in the respective case and which can be further concretized or restricted in the conversation:

“*Brainstem murmur usually occurs before or during a migraine attack and can cause symptoms such as visual disturbances, hearing loss, dizziness, balance problems, numbness or tingling in the face, neck, or extremities, and speech and coordination problems.*”(Case 3.2)

Regarding disease genesis and treatment, treatment approaches were always cited, as well as possible risk factors contributing to disease development in some cases:

“*Several types of medications are available to treat migraine attacks, including analgesics, anti-inflammatories, triptans, and ergotamines. These medications work in different ways and are indicated for different types of migraine attacks.*”(Case 5.1)

The tested language model thus reacted to articulated information needs, but did not anticipate them in advance, whereby objectively determined information needs may not be part of the interactions:

“*To diagnose silent migraine, neurological examinations such as a nervous system examination, an EEG (electroencephalogram), a CT (computed tomography) scan, or an MRI (magnetic resonance imaging) scan may also be performed to rule out other conditions. A blood test may also be performed to ensure that there are no underlying metabolic disorders.*”(Case 6.1)

In addition, advice was occasionally given on preventive measures, mainly related to avoiding risk factors. For example, stress management and relaxation techniques were recommended to prevent stress. On request, the effects of the disease on everyday life were also described, such as an increased risk of anxiety disorders and panic attacks in a migraine with a brainstem murmur. Regarding the category “future perspectives”, ChatGPT addresses preventive measures to avoid migraine attacks and reduce symptoms, considering the people’s fears. Furthermore, the statements made by ChatGPT were checked for the accuracy of the content. Compared with the information sources used as a basis for comparison, the content provided by ChatGPT could be classified as correct.

Building on this, consider the following examples of user queries that demonstrate how varying levels of specificity can impact the responses provided by ChatGPT:

“*Yesterday I was at a party where I drank alcohol and later had problems with the vision in my left eye and sometimes even couldn’t see anything at all. I am 18 years old and female. What could that have been?” In this first example ChatGPT gives the response, that: “It’s possible that the alcohol temporarily affected your optic nerve, which may have caused problems seeing in your left eye.*”(Case 5.2)

The situation was then formulated in the following way:

“*I’m 18 years old and I was at a party recently. During the party I suddenly could hardly see anything in my left eye. I thought I had something in my eye, but it lasted for about an hour. What could that be?*”(Case 5.2)

The given response to that was: 

“*Sudden loss of vision in one eye can have various causes. A possible cause could be a temporary migraine with aura.*”(Case 5.2)

These examples demonstrate the impact of varying levels of specificity in user queries on ChatGPT responses. This means that ChatGPT establishes connections to the given information, which, in the first example, plausibly means that alcohol is a potential trigger for the temporary visual impairment. If this detail is omitted, as mentioned in the second example, connections are made with current, and for ChatGPT, available information, whereby the migraine is named as a probable trigger. In the case of variations in recommendations due to the different specificity and accuracy of the information, users should always provide comprehensive information regarding their disease status. The subjective interpretation of the information to be provided by the user to ChatGPT is problematic. It should be noted that the weighting of relevant health data varies greatly and can therefore lead to different results, as described in case 5.2. As a review of ChatGPT, a wide range of diagnoses or possibilities should be mentioned so that the user is informed about different probable possibilities and the function of ChatGPT, as information generation and decision support, serves the user as a basis for further research. Nevertheless, the contents of the answers of the language model were incongruent. Thus, by performing two rounds per case, the reproducibility of the answers was investigated.

### 3.2. Reproducibility of Information

Similar answers were given in both rounds to questions with identical contents. In most cases, ChatGPT mentioned information in round one that was not mentioned in round two and vice versa, which is why the rounds were classified as non-reproducible. Moreover, in a few cases, the answers contradicted each other between the rounds. For example, in one case, the duration of migraine attacks in round one was reported as four hours, while it was reported as two hours in round two.

Additionally, linguistic comprehensibility was assessed using the Hamburg comprehensibility model [22], which evaluates the categories of simplicity, structure and order, brevity and conciseness, and stimulating additions. Our results approached the optimum in all four categories and can be rated as good to very good (see Table 3). According to the Hamburg comprehensibility model, the values of Optimum, Near Optimum, and Pessimum are defined as follows: Optimum corresponds to the positive range (represented in the model as + or ++), Near Optimum is neutral (represented as 0), and Pessimum falls within the negative range (represented as − or −−).

The evaluations presented in Table 3 indicate that most cases fall within the “Optimum” range (green), with some cases rated as “Near Optimum” (yellow), and none classified as a “Pessimum” (red). These results suggest that the comprehensibility of the responses generated by ChatGPT is generally high, despite the limited reproducibility of the information. Notably, the simplicity category is predominantly rated as yellow, indicating that it is Near Optimum. This suggests that while the information is generally clear, there are areas where simplicity could be improved to enhance overall comprehension.

### 3.3. Situational Appropriateness

Concerning situational appropriateness, the focus was on patient-specific expression, consideration of the emotional level of the affected person and language in general. ChatGPT regularly emphasized the need to consult a doctor, as it was not in a position to make a diagnosis or give medical advice. In addition, the seriousness of the conditions in question was always emphasized:

“*It is important to note that complicated migraine is a serious condition that requires appropriate medical care.*”(Case 9.1)

Regarding the adaptability of ChatGPT to the characteristics of the patients, the answers were inconsistent. The previously mentioned circumstances were often addressed, but essential information was sometimes omitted from subsequent responses. The language of ChatGPT could generally be classified as consistently patient-specific, professional and consistent. Only the address with the polite form “you” was inconsistent within some chat progressions. An exception was case two, which was conducted from a child’s perspective. In this case, ChatGPT nevertheless provided English research in response to the question for further information, which could not be classified as patient-specific. ChatGPT consistently responds understandingly and empathically to stated concerns and anxiety regarding the emotional level:

“*Stay patient and positive and enlist support from friends and family when you need it.*”(Case 3.2)

### 3.4. Information Sources

During the analysis, the information provided by ChatGPT was reviewed to determine whether sources were provided and, if so, whether they existed and were appropriate in terms of content. ChatGPT took different approaches to answering the questions about sources used. On the one hand, ChatGPT provided sources used by itself that were available:

“*Sure, I can give you sources on the information mentioned.*”(Case 8.1)

On the other hand, in eight out of twenty rounds, it did not indicate direct access to specific sources:

“*As an AI model, I rely on a wide range of data sources, including texts from the internet, books, and other written materials. However, I do not have direct access to specific sources or a way to cite a specific source for a specific statement.*”(Case 1.1)

When sources were cited, both when asked for more information and when asked about the sources used by ChatGPT, they predominantly consisted of internet sources. However, most of the sources cited by ChatGPT did not exist at the time of analysis. In addition, the homepages of professional associations and scientific databases were often linked, whereas no links to specific information on the respective types of migraines were provided. Thus, when asked about the sources used by ChatGPT, between zero to ten sources were mentioned, of which, on average, one to two sources existed and were usable in terms of content. Therefore, the information could be classified as less comprehensible and understandable.

## 4. Discussion

The primary aim of this article was to investigate to what extent ChatGPT can be a suitable information source for patients with different migraine manifestations concerning the explanation of the disease, the description of options for action and potential effects on everyday life. Throughout the study, ChatGPT was continuously utilized as an interactive tool. Users described their symptoms and circumstances, which allowed ChatGPT to act as a constant point of contact, suggesting potential causes and providing ongoing support tailored to their migraine experiences.

In the analysis at the content level, the language model names possible diseases that are relevant to the cases examined here. These are different, sometimes after corresponding inquiries or requests for concrete answers. Hints for the diagnosis of a migraine (e.g., neurological examination, blood test) and questions about the effects of the disease on everyday life and the future perspective are only presented by ChatGPT if this was explicitly asked by the patient. Thus, the language model follows a clearly need-oriented communication behavior, which does justice to the individual connectivity of the interaction. However, potential information needs, that the users need more in mind due to different circumstances (e.g., a lack of experience, experienced excessive demands, the perceived complexity of medical contexts), do not flow into the conversation this way. Whereas existing offers of digital health communication via websites may have too little orientation to the needs and target group specificity in the compilation of information [26], the opposite tendency is evident here, which requires patients to be more reflective about their own information needs. At the same time, a targeted formulation of questions can lead to more precise answers. Interaction with the language model thus demands a specific competence from the users, which must be learned in interaction with such AI-based systems and which will expand the complex understanding of digital health competencies in the future. At the same time, interaction with a language model demonstrates the need for the same reflexive competence as is required when searching for information on the Internet: on the one hand, critically questioning the quality of information, and on the other hand, being aware of the limits of one’s knowledge and thus consciously distinguishing oneself from expert knowledge [27].

The recommendations shown in the conversations with ChatGPT are qualitatively equivalent to the standard patient information and can be classified as correct in content. The responses are predominantly appropriate to the situation, as seen in the presentation of empathic reactions. The mostly specifically described circumstances of the examined cases are taken up and thus correspond, at least in part, to the need for the emotional support of the affected persons [12]. The individualization of the interaction experience also shows practical advantages over static Web offerings, as they thus more closely resemble a “natural” counseling situation. Linguistic comprehensibility is also given along the categories of ‘simplicity’, ‘structure and order’, ‘brevity and conciseness’ and ‘stimulating additions’. At the same time, however, only limited reproducibility of the interviews could be established since similar but not identical answers were given to identical questions. Comparable studies, however, describe reproducibility as entirely given [28,29], illustrating the inconsistent study situation on the topic. The background of these different study results can be explained by variations in the study methods and the use of different assessment and evaluation methods.

In the study by Yeo et al. [28], the accuracy of the information given by ChatGPT was investigated in addition to the reproducibility. The information given by the AI was found to be 76.9% accurate in the context of quality [28]. However, other studies regarding the quality of the generated information differ substantially from the relatively positive findings shown here [13,30]. For example, Moskatel and Zhang [30] concluded in their qualitative study of using ChatGPT to evaluate the efficacy of medications for migraine prevention that the responses were tailored, but many undetectable cited sources made them unreliable and inaccurate [30]. On the one hand, this could be due to the function of ChatGPT where word associations are created based on probabilities, resulting in content variance in the generated texts, which makes it difficult to compare studies [13]. On the other hand, the quality of the outputs is conditioned by the quality of the training data and the possible biases that the training data hold [31]. The present analysis is congruent with the findings of a lack of traceability, which can be attributed to the limitation, as mentioned above, in the timeliness of the training data [13]. At this point, the possibility must be considered that the Internet-based sources, classified as non-existent at the time of data collection (05 April 2023), may have existed when the data basis or the training data of the language model was compiled (09/2021). The evidence presented here paints a comparatively positive usage experience in disease-related interactions with ChatGPT from the patient’s perspective. However, the partially faulty function of ChatGPT and the questionable application of AI in clinical practice should not be neglected. Due to the given uncertainties regarding the use of ChatGPT in such settings and the multiple factors underlying this characteristic, it is not sufficient to use only the coherence of responses as a quality indicator of ChatGPT to answer the main criteria of this study. Rather, other factors should be added to evaluate the functioning of ChatGPT in a medical setting to gain a more comprehensive insight into the use and information processing of ChatGPT.

Previous research approaches thereby support the currently prevailing inconsistent study situation regarding using ChatGPT in obtaining patient information. Study results suggest that there are isolated differences between the utterances made by ChatGPT and those of Healthcare experts, but the information cited by ChatGPT was accurate and of consistent quality [14]. Other case studies show a lack of scientific accuracy and emphasize the limited knowledge and lack of ability to discuss results critically [32].

The study results are subject to various limitations from which further research needs can be derived. It should be particularly emphasized that ChatGPT is not a scientific source. Thus, information may also be invented or misrepresented by the system. The use of ChatGPT in a health-related context also requires digital health skills, which help to filter and evaluate relevant information. The individual assessment of this largely determines the quality of the further responses of the AI system and the resulting benefit for the patient. A bias or influence of the answers, e.g., an expectation of the interviewers formed by the interest in knowledge, cannot be excluded because no external interviewers were used. The transferability to the use of other AI-supported language models is limited due to the different modes of operation. The same applies to considering the diversity-sensitive information needs of different target groups, which were only partially incorporated into the cases considered. Furthermore, due to the dynamic technological development, the findings here can only be regarded as snapshots. Against this background, there is a need for continuous scientific monitoring of the use of AI-based systems in health communication, particularly regarding the effects achieved, which may differ from previous digital health communication due to the target group specificity, the potential need for orientation and the empathy shown. Also, consumer protection, data and personality protection, equal opportunities and the future influence of economic interests on the design of content as well as the profiling of user groups as a business model, a special consideration of the potential and risks of privately used technologies such as ChatGPT, are required, especially against the background of their application in the context of health [33].

The findings need to be assessed taking into account some limitations. First, the study relied on fictitious cases based on typical migraine characteristics. While this approach allows for controlled analysis, it may not fully capture the diversity and complexity of real-world patient experiences. Additionally, the fictitious nature of the cases may limit the external validity of the findings and therefore cannot be applied to clinical use cases. A significant limitation is the dependence on ChatGPT version 3.5. The conversations conducted in this study utilized this version, which may have limitations in understanding and responding to complex medical queries. The model’s knowledge is based on data available up to its last training cut-off in September 2021, potentially limiting its access to the most current medical information and developments. Future iterations of the model might provide more accurate and updated responses, but the current findings are constrained by the capabilities of version 3.5. Another major limitation was the verification of sources provided by ChatGPT. In many instances, the sources cited did not exist or were too general, often linking to homepages rather than specific articles or data relevant to the query. This lack of verifiable sources undermines the credibility of the information provided and poses a challenge for users seeking reliable and traceable health information. The study’s scope was limited to migraine and tension headaches, which may not be representative of other headache disorders or broader health conditions. This narrow focus limits the generalizability of the findings to other medical contexts. Conducting semi-structured interviews with an AI model raises ethical considerations, particularly regarding the model’s limitations in providing medical advice. Users must be aware that ChatGPT cannot replace professional medical consultation and that its suggestions should not be used as a basis for self-diagnosis or treatment.

In the future, it will be necessary to show whether such services are perceived as a sufficiently low threshold to address hard-to-reach target groups in health communication (e.g., people with migration and refugee experiences, people with disabilities, people who are very old, or people from educationally disadvantaged backgrounds) [34]. Other areas of use for such applications should also be tested, e.g., translating medical terms, summarizing articles in simple language for people with cognitive impairments, translating health information into other languages, and whether such applications can contribute to diversity-sensitive and culturally sensitive health communication [35]. It will be crucial that the development is oriented towards a consistent user orientation, that prevention and care services are complemented meaningfully, and that existing technological innovations are evaluated with the help of relevant health endpoints. At the same time, technology’s social, ethical, and health-related consequences must be examined in an interdisciplinary manner, which requires funding independent of industry. In addition, the latest versions of ChatGPT, ChatGPT-4 and ChatGPT-4o, provide new research avenues for the future use of ChatGPT as a diagnostic tool and information generator for migraine patients. These versions can be examined considering the main features studied, as well as their temporal development. The advanced capabilities of ChatGPT-4 and ChatGPT-4o offer improved accuracy and enhanced contextual understanding, which can potentially address some of the limitations noted in previous versions. These updates could lead to more precise identification of migraine types, better-tailored management strategies, and more reliable patient support. Future research should focus on evaluating the effectiveness of these newer versions in clinical settings, assessing their impact on patient outcomes, and exploring their integration with existing healthcare systems to optimize migraine care.

## 5. Conclusions

While this study aimed to explore ChatGPT’s potential as an information source for patients with migraines, the findings reveal that its use in this context lacks substantial research value due to inherent limitations in the model’s application. The necessity for users to explicitly request specific information, combined with the variability in response accuracy and reproducibility, underscores the model’s dependency on user proficiency in digital health literacy. These factors limit ChatGPT’s suitability as a standalone tool for comprehensive patient support.

Instead of focusing solely on ChatGPT’s capabilities, future research should explore broader implications and alternative directions. This includes investigating the integration of AI tools with existing healthcare systems to enhance patient care, examining the ethical and social impacts of AI in health communication, and developing strategies to improve digital health literacy among users. Additionally, assessing the effectiveness of newer AI models like ChatGPT-4 in clinical settings could provide deeper insights into their potential benefits and limitations.

In summary, while ChatGPT demonstrates some promise, its current application in migraine patient support is limited. Further interdisciplinary research is essential to optimize the use of AI in healthcare and to address the diverse needs of patient populations effectively.

## Figures and Tables

**Table 1 healthcare-12-01594-t001:** Description of the fictitious cases.

Number	Type of Migraine	Characteristics of the Cases	Confirmed Diagnosis
1	Migraine attacks	Female (30 years), stressful life, increased alcohol consumption, odor sensitivity, 2–3 h migraine attack without aura.	No
2	Familial hemiplegic migraine	Male (13 years), father has hemiplegic migraine, prolonged headache, visual disturbances, hemifacial paralysis, hypoesthesia.	No
3	Migraine with brainstem murmur	Male (29 years), severe headache, dizziness, impaired consciousness, rapid heartbeat, panic attacks.	Yes
4	Eye migraine	Female (14 years), hormonal changes, first experience with alcohol, stress, accompanied by headaches, visual disturbances and eyestrain (flickering, double vision, visual field impairment).	No
5	Retinal migraine	Female (18 years), slightly overweight, hormonal contraception, alcohol and nicotine consumption, sleeps little and has an unhealthy diet, visual disturbances in one eye, flickering and temporary blindness.	No
6	Typical aura without headache	Female (27 years), hormonal fluctuations due to pregnancy, reversible visual aura, sees black dots or double, visual field loss and metamorphopsia, dizziness, tingling and numbness in extremities.	No
7	Migraine with prolonged aura	Female (33 years), hemiplegic migraine has already been diagnosed, known triggers are bright light and stress, hormonal fluctuations, motor symptoms and sensory symptoms of aura, which usually last for a few days	No
8	Migraine infarct	Female (37 years), hormonal contraception, nicotine use, treatment with triptans, more severe paralysis (face and left arm), visual disturbances, mild speech disturbances, balance disturbances (symptoms occur simultaneously).	Yes
9	Status migrainosus	Female (28 years), migraine with aura in early teens, stress, stopped hormonal contraception, changing symptoms: severe unilateral headache lasting at least 72 h, nausea, sensitivity to noise and light, visual field impairment, speech disorders.	no
10	Tension headache	Male (35 years), stress, pressure, anxiety, bilateral headache gradually rising from neck to forehead, vice-like pain.	No

**Table 2 healthcare-12-01594-t002:** Structure of the interview guide.

Thematic Block	Content	Example
Definition (patients with a diagnosis)	Specific diagnosis and accurate description of the symptoms.	I have […]. What disease could this be?
Anamnesis (patients without a diagnosis) ^b^	Presentation, description of symptoms, situations and potential triggers.	My name is […]. I am […] years old and suffer from the following symptoms: […].
Therapy (patients with diagnosis)	Queries/requests about disease and potential triggers.	What other symptoms may occur?
Possible diagnoses ^b^ (patients without a diagnosis)	Specifying symptoms and probability of the disease.	Are there (other) diseases that match my symptoms?
Instructions for action (patients with a diagnosis) ^b^	Prevention, advice on seeking out the healthcare system and self-help.	How can I act preventively? How can I relieve symptoms myself? Are there any self-help groups?
Instructions for action (patients without a diagnosis)	Choice of doctor, immediate recommendation of action, procedure of diagnosis and treatment options.	Which doctor should I consult? How will diagnosis proceed? What are treatment options?
Future perspective/ essential questions	Research on cure, gain in health-related quality of life, heredity, impact on everyday life.	Can my children be affected by the disease, too? How far advanced is research into the curing of migraines?
Sources	Further information, sources for the recommendation.	Where can I get more information? Can you give me the sources where you found the information?

Note: All topic blocks are marked with a reference to cases and diagnosis, those marked with “^b^” refer to cases without a diagnosis, sample questions.

**Table 3 healthcare-12-01594-t003:** Summarized evaluation of the two rounds based on the Hamburg comprehensibility model [22], differentiated by case and category.

Case	Simplicity	Structure and Order	Brevity and Conciseness	StimulatingAdditions
1				
2				
3				
4				
5				
6				
7				
8				
9				
10				


 Optimum. 
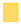
 Near Optimum.

## Data Availability

The datasets analyzed during the current study are available from the corresponding authors upon reasonable request.

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
