# Peer review of "ChatGPT as an Information Source for Patients with Migraines: A Qualitative Case Study"

_healthcare, 2024, doi:10.3390/healthcare12161594_

Round 1
Reviewer 1 Report
Comments and Suggestions for Authors
The authors present qualitative case study using ChatGPT as an information source for patients with migraine. The paper is well written covering the details of the study along with the discussion on results as well as the limitations. Some points need to be addressed to further improve the article.
The abstract does not convey any information that why the study is needed. What are the problems that patients with migraine face? How do the patients get such information if they do not use ChatGPT?
The introduction section should also cover the research gap and the need for the study. Why ChatGPT is selected for the study? What are the tools other than ChatGPT?
At the end of the introduction section, the structure of the paper needs to be included.
The scenario discussed in lines 215-239 presents a significant challenge for the users. How the users understand the specificity and the need to change the input. The results presented do not cover any possible remedies for such situations.
In Table 3, the “pessimum” does not exist. Either there is error in the table or the extra item from the legend needs to be removed.
Author Response
Comment 1: The authors present qualitative case study using ChatGPT as an information source for patients with migraine. The paper is well written covering the details of the study along with the discussion on results as well as the limitations. Some points need to be addressed to further improve the article.
Response 1: Firstly, we would like to express our gratitude for your review of our manuscript and the constructive feedback you have provided. We have addressed these comments in our revisions and have striven to implement them as effectively as possible.
Comment 2: The abstract does not convey any information that why the study is needed. What are the problems that patients with migraine face? How do the patients get such information if they do not use ChatGPT?
Response 2: Thank you for this hint. We have revised the abstract with the aim of providing a more precise description of the added value of the study (see lines 9-17).
Comment 3: The introduction section should also cover the research gap and the need for the study. Why ChatGPT is selected for the study? What are the tools other than ChatGPT?
Response 3: Thank you for the comment. We gave here more explanation to clarify the research gap and the relevance of our study (see lines 80-88).
Comment 4: At the end of the introduction section, the structure of the paper needs to be included.
Response 4: We here included the structuring of the paper:
In the following, first the methodological approach of the analysis is explained. For this the research object and methods of data analysis are described in detail. Afterwards, the results of the content analysis and discussion of the insights as well as the limitations of the study are presented (see lines 116-121).
Comment 5: The scenario discussed in lines 215-239 presents a significant challenge for the users. How the users understand the specificity and the need to change the input. The results presented do not cover any possible remedies for such situations.
Response 5: Thank you very much for the feedback. We here added further details and possible remedies for this (see lines 256-264).
Comment 6: In Table 3, the “pessimum” does not exist. Either there is error in the table or the extra item from the legend needs to be removed.
Response 6: Thank you for the hint. We removed the option “pessimum” in the legend (see table 3).
Reviewer 2 Report
Comments and Suggestions for Authors
In this paper, the authors portend to analyze ChatGPT’s ability to diagnose migraine in a variety of scenarios. They submit the scenarios to ChatGPT and analyze the answers along several dimensions. Their results suggest quality answers, except for the lack of traceability/source of the answers.
The paper is well written and addresses a critical aspect of future medical consultations, namely the capacity of chat agents to provide good answers.
My main concern is the lack of documentation. As a reviewer, I would like to see the entirety of the interview guide and much more detail in terms of how the responses were coded. While I understand that the method is qualitative, the authors derive fairly rigid ordinal categorizations (“optimum”, “near optimum”…), on which they base their discussion. As a consequence, it is important to present this analytical process at a level of detail that enables reproducibility.
Author Response
Comment 1: In this paper, the authors portend to analyze ChatGPT’s ability to diagnose migraine in a variety of scenarios. They submit the scenarios to ChatGPT and analyze the answers along several dimensions. Their results suggest quality answers, except for the lack of traceability/source of the answers.
Response 1: Firstly, we would like to express our gratitude for your review of our manuscript and the constructive feedback you have provided. We have addressed these comments in our revisions and have striven to implement them as effectively as possible.
Comment 2: The paper is well written and addresses a critical aspect of future medical consultations, namely the capacity of chat agents to provide good answers.
Response 2: Thank you very much for your comment on our paper.
Comment 3: My main concern is the lack of documentation. As a reviewer, I would like to see the entirety of the interview guide and much more detail in terms of how the responses were coded. While I understand that the method is qualitative, the authors derive fairly rigid ordinal categorizations (“optimum”, “near optimum”…), on which they base their discussion. As a consequence, it is important to present this analytical process at a level of detail that enables reproducibility.
Response 3: Thank you very much for your feedback. We revised the methods section in terms to be more comprehensible and traceable, now.
Reviewer 3 Report
Comments and Suggestions for Authors
MINOR REVISION In this paper, a consistency criterion was applied to the answers and suggestions in migraine sufferers' health counseling. The recommendations made by ChatGPT have been examined to see if they are qualitatively comparable to conventional patient information, accurate in content, comprehensible in language, suitable for the circumstance, and appropriate. The degree to which individualized demands resulting from modifications in the disease's trajectory were considered was examined. In the experimental cases, ten distinct forms of migraines were identified, along with their associated consequences, and ten hypothetical instances with characteristic symptoms were created, each with a specific type of migraine or complication. They structured an interview guide with eight queries for different themes. Content analysis, Reproducibility of information, Situational appropriateness, and Information sources. The paper is informative and has interest for readers. Although some notable research has been done in the study, its findings are not satisfactory enough. The findings need to be presented more carefully. The amateur findings given in the conclusion, "ChatGPT gives good comments, but use it with caution", should be presented more technically.
Comments on the Quality of English Language
Minor editing of English language required
Author Response
Comment 1: In this paper, a consistency criterion was applied to the answers and suggestions in migraine sufferers' health counseling. The recommendations made by ChatGPT have been examined to see if they are qualitatively comparable to conventional patient information, accurate in content, comprehensible in language, suitable for the circumstance, and appropriate. The degree to which individualized demands resulting from modifications in the disease's trajectory were considered was examined. In the experimental cases, ten distinct forms of migraines were identified, along with their associated consequences, and ten hypothetical instances with characteristic symptoms were created, each with a specific type of migraine or complication. They structured an interview guide with eight queries for different themes. Content analysis, Reproducibility of information, Situational appropriateness, and Information sources. The paper is informative and has interest for readers. Although some notable research has been done in the study, its findings are not satisfactory enough. The findings need to be presented more carefully. The amateur findings given in the conclusion, "ChatGPT gives good comments, but use it with caution", should be presented more technically.
Response 1: Firstly, we would like to express our gratitude for your review of our manuscript and the constructive feedback you have provided. We have addressed these comments in our revisions and have striven to implement them as effectively as possible.
Reviewer 4 Report
Comments and Suggestions for Authors
This manuscript interviews ChatGPT on a specific topic, which has some exploratory significance and is a very interesting study.
As we all know, ChatGPT's answers are not scientifically meaningful and should not be used in clinical practice.The text generated by GPT is likely to have logical problems or factual errors. From a technical point of view, it is not appropriate for this study to take answering the question or whether the scheme given is consistent as the main research goal, because this itself is difficult to avoid in the general model technology.
At present, ChatGPT4 and 4o have been widely used, and their performance has obviously exceeded ChatGPT3.5. We suggest the author increase the comparison with GPT4. In this way, the direction of evolution of AI technology can be better understood.
The conclusion given by the author is very rigorous, but the conclusion itself has been declared in the use of GPT, so the conclusion of this manuscript is not of research value, and the author is advised to summarize from other directions.
Author Response
Comment 1: This manuscript interviews ChatGPT on a specific topic, which has some exploratory significance and is a very interesting study.
Response 1: Firstly, we would like to express our gratitude for your review of our manuscript and the constructive feedback you have provided.
Comment 2: As we all know, ChatGPT's answers are not scientifically meaningful and should not be used in clinical practice. The text generated by GPT is likely to have logical problems or factual errors. From a technical point of view, it is not appropriate for this study to take answering the question or whether the scheme given is consistent as the main research goal, because this itself is difficult to avoid in the general model technology.
Response 2: Thank you very much for your comment on our paper. We have included this aspect in the discussion of our results (see lines 411-418).
Comment 3: At present, ChatGPT4 and 4o have been widely used, and their performance has obviously exceeded ChatGPT3.5. We suggest the author increase the comparison with GPT4. In this way, the direction of evolution of AI technology can be better understood.
Response 3:
Thank you very much for this interesting aspect. We considered this in our discussion section and added this as a possible idea for future research (see line 461-471).
Comment 4: The conclusion given by the author is very rigorous, but the conclusion itself has been declared in the use of GPT, so the conclusion of this manuscript is not of research value, and the author is advised to summarize from other directions.
Response 4: Thank you very much for the feedback. We revised the conclusion to be more precise now.